# Semi-Autonomous Vehicles as a Cognitive Assistive Device for Older Adults

**DOI:** 10.3390/geriatrics4040063

**Published:** 2019-11-16

**Authors:** Frank Knoefel, Bruce Wallace, Rafik Goubran, Iman Sabra, Shawn Marshall

**Affiliations:** 1Bruyère Memory Program, Ottawa, ON K1N 5C8, Canada; 2Bruyère Research Institute, Ottawa, ON K1N 5C8, Canada; wally@sce.carleton.ca (B.W.); goubran@sce.carleton.ca (R.G.); isabra@bruyere.org (I.S.); 3Faculty of Medicine, University of Ottawa, Ottawa, ON K1H 8L6, Canada; smarshall@toh.ca; 4Department of Systems and Computer Engineering, Faculty of Engineering and Design, Carleton University, Ottawa, ON K1S 5B6, Canada; 5AGE-WELL NIH—SAM^3^, Ottawa, ON K1N 5C8, Canada; 6Ottawa Health Research Institute, The Ottawa Hospital, Ottawa, ON K1Y 4E9, Canada

**Keywords:** aging drivers, older drivers, driving impairment, driving cessation, adapting to semi-autonomous vehicles

## Abstract

Losing the capacity to drive due to age-related cognitive decline can have a detrimental impact on the daily life functioning of older adults living alone and in remote areas. Semi-autonomous vehicles (SAVs) could have the potential to preserve driving independence of this population with high health needs. This paper explores if SAVs could be used as a cognitive assistive device for older aging drivers with cognitive challenges. We illustrate the impact of age-related changes of cognitive functions on driving capacity. Furthermore, following an overview on the current state of SAVs, we propose a model for connecting cognitive health needs of older drivers to SAVs. The model demonstrates the connections between cognitive changes experienced by aging drivers, their impact on actual driving, car sensors’ features, and vehicle automation. Finally, we present challenges that should be considered when using the constantly changing smart vehicle technology, adapting it to aging drivers and vice versa. This paper sheds light on age-related cognitive characteristics that should be considered when developing future SAVs manufacturing policies which may potentially help decrease the impact of cognitive change on older adult drivers.

## 1. Impact of Aging on Driving

Driving is a complex instrumental activity of daily living requiring cognition and good physical health. With the increasing numbers of older drivers driving more miles than ever before, it is important to be conscious of age-related declines in sensory, cognitive, and psychomotor abilities that can potentially affect older drivers on the road [1]. This paper will focus on cognitive change and explore how the development of semi-autonomous vehicles (SAVs) with consideration of the older driver could delay older adult driving retirement.

There is general agreement that with aging there is a complex interaction of genetics, life experience, and lifestyle, which ultimately result in the cascade of intracellular changes, then cell atrophy and ultimately death; cognition is impacted directly by the resulting decline in inter-cellular communication and neural network functioning [2,3,4]. Many cognitive domains can be impacted by aging, but for the purpose of creating a model, this paper will focus on visual scanning, attention, speed of processing, and some components of executive functioning and memory, as these domains have repeatedly been shown to be related specifically to driving performance in older adults [5,6].

Visual scanning is essential for driving. Much in the environment needs to be monitored visually, including: where a person finds themselves, other vehicles, potential road obstacles, changing traffic lights, and information about their own vehicle: speedometer, lane change indicators, notifications on vehicle requirements (amount of gas available, required maintenance). All of these parameters change continuously while driving and hence visual scanning is essential for safe driving. A number of papers have shown that visual scanning ability declines with age [7,8]. In terms of driving performance, it was noted that older drivers scanned significantly less toward the left and right during intersection negotiations compared to middle-aged and younger drivers. Older drivers also focused more on one traffic stream before executing a right or left turn and had significantly fewer glances toward the turning direction. Ultimately this results in them having a greater likelihood of car crashes involving a right or left turn movement [9].

With all this visual and other information carried into the brain, it needs to be able to focus on selective, relevant information to make decisions around its importance on the current driving task and ignore other less relevant stimuli. Selective attention is therefore required. Research on attention has shown a number of changes associated with aging [5,10]. Inattention to driving performance can result in degraded lane keeping, poor speed control, delayed reaction time, missed traffic signals and road signs, shorter or longer following distances, unsafe gaps, reduced situational awareness, poorer visual scanning [11], and poorer target awareness [8]. Naturalistic driving research has demonstrated that drivers with higher driving inattention are more likely to be involved in car crashes [12,13].

Speed of processing is the brain functioning component affecting response time. With regard to driving, the visually scanning brain will ideally attend to an emerging image in the periphery long enough to identify it as, for example, a child on a bicycle coming down a driveway, allowing the driver to make the appropriate changes in to speed and direction quickly to avoid an accident. There is evidence that speed of processing slows with aging [14]. It was found that older adults take 1.7 to 2.0 times longer than younger adults to process and respond to basic information related to cognitive, perceptual, and motor processing operations [15]. Studies have shown the impact of reduced speed of processing on driving. Older drivers with a decline in information speed of processing are more than twice as likely as older adults with intact speed of processing to sustain an at-fault crash [14,16].

Executive functions are a plethora of psychological functions that include elements such as dividing attention, cognitive flexibility, inhibition, planning, reasoning, and self-regulation. All elements of driving require executive functioning: from planning the trip and adapting the plan based on reality, to second to second decisions and evaluating the impact of those decisions. This is emphasized in Michon’s Hierarchical model of driving, as well as in Anstey’s behavioural model [6,17]. Aging has been shown to affect elements of executive function in part compounded by the reduced ability to process information quickly. Consequently, a driver’s capability to divide attention between a dominant location and its surrounding, as well as rapidly switch attention degrades significantly with age [14,18]. Driving research has shown that a decline in executive functioning can include a reduction in the speed at which visual information is processed when switching between two aspects of driving, and that this can lead to potentially dangerous traffic situations. An example would be when driving on the road where the driver is looking in the rear-view mirror to make a lane change, and then switching attention to the front to detect a tailgating situation with the lead vehicle.

Finally, memory declines with age for normally aging older adults, as well as for those affected by mild cognitive impairment and dementia. Approximately 40 per cent of individuals over the age of 65 experience some type of memory loss [19]. Interestingly, memory decline by itself seems to have less of an impact on driving ability, because most driving skills are overlearned, and other cognitive domain changes seem to impact driving safety before memory does. 

There is abundant evidence that collectively these declines in visual scanning, selective attention, speed of processing, and executive functioning have an impact on actual driving outcomes. Older adults are more likely to engage in unsafe left-hand turns across opposing traffic at intersections [20], have difficulty in navigating, and respond inappropriately to rapid altering road situations [14,18]. A review of over 5000 older adult driver crashes in Florida, found that lane maintenance, yielding and gap acceptance errors predicted crash-related injuries with almost 50% probability [21].

Given all of these cognitive changes associated with aging, theoretically over time potentially every driver would develop changes sufficient to increase their crash rate, while recognizing that this is likely balanced out by experience and changes in behaviour such as limiting driving exposure and changes in driving style. And hence crash risk would be even more amplified in persons with certain types of neuro-degeneration. For instance, cognitive decline associated with dementia of Alzheimer’s type, has been shown to increase the number of intersection errors, speed errors, problems with lateral control, and crashes on driving simulators [17,18,19].

Clinicians in many jurisdictions have legal obligations to report drivers at an increased risk of car crash, but even in jurisdictions where there is no law, they likely have moral obligations to the safety of older drivers and others, including family. The current practice to assess driver risk includes clinic-based cognitive assessments and on-road driving tests. The literature is mixed regarding the correlation between cognitive tests and actual on road driving [5,22] and on-road tests remain the gold standards. At this time, doing poorly on in-clinic or on-road testing can lead to attempts at driver training or rehabilitation. There have been successes in driver training both on-road and using a driving simulator [23]. However, if these are unsuccessful, license suspension and driving retirement usually follow. The legal and ethical issues around driving retirement or its delay are significant.

Nevertheless, the impact of removing an older adult driver’s license can lead to significant declines in health and social functioning. Numerous papers have explored the negative effects the loss of driver’s license has had on older drivers’ general health, mental state, and social well-being. First of all, there can be a direct impact on social connectedness with friends and relatives [24], or a loss of productive engagement in volunteering and employment [25], leading to feelings of loss of independence and control over one’s life [26,27]. This is turn can lead to depressive symptoms [28] and even cognitive decline [29]. Finally, there can be an additional overall decline in general health [5], higher risk admission rates into long-term care facilities [28], and even death [29].

## 2. Vehicle Autonomy

Given that driving cessation can result in significant implications for older adults, it is crucial to explore opportunities to safely extend their driving. The research into fully autonomous vehicles shows promise for the future and may extend local mobility beyond the time of having an active driver’s license for aging drivers. However, we are currently in a period of time that is advancing towards this goal. This section explores the current state of vehicle autonomy.

The US Department of Transportation (USDOT) has determined that there are five levels of vehicle automation, which are described in Table 1 [30].

Automation requires a number of sensors installed in and around the vehicle to continuously observe their surrounding environment and accurately calculate their location in a lane, the nearby vehicles and curb, obstacles, intersections, and vehicle speed [31]. These sensors allow the control of vehicle dynamics with the objective of following the trajectory requested by the human driver in the best possible way [32]. Automotive sensing falls into three main categories as described in Table 2 [31,33].

In addition to sensing, driverless automation requires new traffic control infrastructure (traffic lights/signs) that wirelessly communicate with the car. This information must then be integrated by the software that makes decisions about driving safety and ultimately gives direction to the vehicle. 

### 2.1. Current Semi-Autonomous Vehivle (SAV) Features

Current commercially available SAVs include features like: adaptive cruise control (ACC), lane departure warnings, collision avoidance, parking assist systems, and on-board navigation [34]. As examples, Toyota cars have a Lane Trace Assist that uses a camera and radar to help keep the car in its lane; Cadillac’s Super Cruise uses a camera to track the driver’s head position. If the driver looks away from the road for more than a few seconds, they receive an audio and visual warning, along with a seat vibration notice [35].

Higher-end car manufacturers have explored the possibility of including sensor systems in seat belts and seat covers of cars that intervene in case of deficient driving abilities for example, the detection of driver fatigue and drowsiness. These systems are based on drivers’ steering behaviour and response times, length of the trip, use of turn signals, and time of day [32]. For instance, BMW is investigating the incorporation of steering wheel sensors that monitor vital signs and stress levels using metrics such as blood-sugar monitoring, heart rate, and blood oxygen saturation, and take measures to reduce distractions or slow down the vehicle [36]. Mercedes-Benz is exploring the addition of steering wheel sensors to detect inconsistent driving behaviour. Finally, Lexus is proposing to use cameras to watch for drowsiness [37].

Additional beneficial technology includes systems that enhance night vision such as intelligent headlights [38]. This technology permits lighting of specific angles and areas of relevance for a short time to attract the driver’s attention and maximize the illumination support of the driver through a distributed antenna system. Other night vision systems are based on close-range and long-range infrared technology, providing an on-screen display of the driver’s surrounding object environment (e.g., pedestrians, bikers) and thus trigger a warning notification to the driver [32].

### 2.2. Technology Benefits

The public, government agencies, health care providers, and researchers are excited about the prospect of vehicle automation to improve road safety while preserving driving autonomy. Vehicle automation has the capacity to significantly reduce travel times, road accidents, and congestion [38]. According to the National Highway Traffic Safety Administration [39], human error and human factors such as inattention, distraction, or speeding are thought to be responsible for over 90% of all accidents [40]. Moreover, a third of accidents and mortalities could be avoided if vehicles have automation options such as forward collision and lane departure warning systems, side view (blind spot) assist, automatic braking, and adaptive headlights [41]. Road congestion can be reduced with AVs using existing lanes and intersections more competently via shorter gaps between vehicles, coordinated platoons, and selection of efficient route choices [34].

This technology can also potentially have great ecological benefits by reducing fuel consumption and greenhouse gas emissions from the capability to deploy vehicles according to each trip’s occupancy (right-sizing) [42]. Fuel economy has shown to be improved by 4–10 percent when SAV technologies accelerate and decelerate more smoothly than a human driver [43]. Furthermore, SAVs can sense and possibly anticipate a front vehicles’ breaking and acceleration reaction. Thus, the SAVs can break smoothly and make appropriate speed adjustments, resulting in fuel savings, less brake wear, and reductions in traffic-destabilizing shockwave propagation [34].

It is possible to conceive of a future where a vehicle with automation could be considered an assistive device like a walker or a hearing aid. Furthermore, we imagine that future driver’s license categories might include vehicle automation, just like a current driver’s license requires eyeglasses be worn by certain drivers. Moreover, one can conceive of a future where driving automation is an assistive device for people with cognitive challenges delaying driving retirement. In a similar way that glasses aid in visual clarity, and walkers support walking independence, driving automation could help decrease the impact of cognitive change on older adult driving.

## 3. Proposed Model of How to Connect Needs of OA to SAV

This section explores the potential of using SAVs as a cognitive assisted device for older adults. Figure 1 gives a sense of the complexity of a model that connects driving changes associated with cognitive decline and vehicle automation. The far-left column shows some of the cognitive domains that change with age and that can impact driving ability. These can be mapped onto the second column which shows frequent impacts of these cognitive changes on actual driving behaviour. The third column shows many of the sensing technologies currently available in vehicles. Finally, the far-right column shows currently available automation elements. 

Figure 2 shows how the model could be used. Lane drifting is provided as an example. Cognitive domains associated with aging that could contribute to lane drifting include visual scanning, attention, speed of processing, and executive functioning. Of the various sensors currently available, the lane position sensors would be the most relevant to detect lane drifting. A smart driving system could alert the driver about drift, e.g., through a shaking steering wheel, visual cue or sound, and with further automation the steering could be corrected directly.

## 4. Human Computer Interaction

As seen above, SAVs are equipped with numerous forms of sensor and stimuli features that share tasks and responsibilities with human drivers. This human computer interaction (HCI) allows for a safe and comfortable driving experience. It involves the human driver and their potentially health-compromised factors, the AV features that supplement them and the level of assistance provided in the particular driving situation [1]. The current Level 1 SAV assumption is that the human driver is in charge and is only assisted by the AV system whenever there is an unexpected hazard providing alerts to the driver such as blind spot detection. With the transition to Level 2 through 4 SAVs, the HCI model dramatically shifts where now the driver is supervising the SAV features and is expected to intervene when the SAV is not able to handle a situation. In Level 2 and 3, the SAV requires that the driver take over in these cases while in Level 4, should the driver not take over, the system automatically helps stop the vehicle or pilots it to a safe position [32]. Level 1 SAV technology features such as blind sport detection are considered as the automobile helping the human driver, same as the sound and light warnings for lane drifting. However, as for the SAV preventing a collision for an unsafe lane change is a very new idea as the assumed model for SAV technology development has focused on a capable driver with a goal of the SAV technology becoming increasingly in charge and not from a perspective of it increasingly supporting a driver with their driving. Nevertheless, with such beneficial features come many challenges that will be explored in the subsequent section.

## 5. Challenges

As seen above, automation brings forward many benefits for the consumers and environment. Nevertheless, there are important challenges to be considered with this technology. These challenges are related directly with the SAV technology itself, older drivers adapting to SAV, and hence how to design senior-friendly SAV technology for the future. 

### 5.1. Technology

#### 5.1.1. Vehicle Costs

The first challenge faced by older drivers/consumers is the cost of SAVs. Currently most commercially available vehicles only include level 1 to level 2 autonomy and the technology for level 3 and beyond vehicles is not mature enough and hence not approved for on-road use. In addition, high costs are predicted for the sensors, communication and guidance technology, and software in level 3 to level 5 AVs [31]. Most of current Level 2 SAVs cost well over $50,000 (US currency) each, and additional costs will accumulate if other features are added such as: additional sensors, high beam assist, active lane assist, forward collision avoidance, adaptive cruise control, top view camera, and autonomous navigation system [34]. Furthermore, autonomous driving systems need to be maintained, updated, and repaired by highly trained automotive specialists, similar to aviation service standards, further increasing costs [44]. Thus, in their current iteration, they are typically unaffordable to the lay older population. In fact, while the literature suggests that older drivers are interested in having SAVs, the majority are not willing to pay extra for the technology [45].

#### 5.1.2. Safety

Safety is a second challenge auto manufacturers face. They need to design systems that can perform safely and handle virtually every possible environmental situation [46]. For instance, the problem of recognition of the wide variation in humans and other objects and reaction to circumstances appearing on the road is problematic for SAV sensors [47]. Sensors and driving operations face challenges related to changing and poor weather conditions such as fog, snow, ice, and heavy rain. Pedestrians on the road that may be out of sight, and who could be standing, walking, sitting, also complicate SAV sensor recognition [34]. Because of these challenges, level 3 automation and beyond have not yet been approved for use on the road by regulators.

Recent accidents have initiated concerns regarding the drivers’ understanding and capability of safely using the technology [31]. Furthermore, the recent examples of the Tesla and the 737 MAX 8 crashes suggest that smart vehicle systems are not sufficiently reliable at this time to allow full automation [48]. Accumulating evidence suggests that the flight automation in the Boeing 737 Max 8 jets (Lion Air 610 and Ethiopian Airlines 302) contributed to the crashes, despite the best efforts of the crew [49,50] even though flight automation has been in use for significantly longer than AVs. Thus, more work needs to be done to fully understand the safety of the human/SAV interaction before driving automation can become a reality. Moreover, there are concerns of threats to electronic security where computer hackers, disgruntled employees, terrorist organizations, and/or hostile nations, may manipulate SAV systems to intentionally cause collisions and traffic disruptions.

#### 5.1.3. Litigation, Liability, and Ethics

Safety concerns bring on questions of insurance and liability. Although SAVs are equipped with sensors, visual interpretation software, and algorithms that allow them to make informed decisions, there may be occurrences where a crash is unavoidable. An example of such situation could be, if an animal jumps in front of the car. In this case, what would be considered the appropriate reaction: does the SAV hit the animal or run off the road to avoid it? How would this reaction change if, instead of the deer, it was another car, a cyclist, or pedestrian? Who would be at fault for the SAV lane departure reaction resulting in striking another vehicle or pedestrian? Such reactions may be questioned in a court lawsuit [34], as well as challenged by public opinions and beliefs. Additional legal issues can arise if there are disagreements between physicians, families, and older drivers regarding the need of driving automation—if there is an accident that could have been avoided using automation, who is responsible? Similarly, if there is an accident despite automation, who is legally responsible: the physician that “prescribed” it, the driver, or the manufacturer? All of these issues will need to be taken into consideration when discussing the risks and benefits with patients and families of choosing to continue to drive with some driving automation vs. driving in retirement.

#### 5.1.4. Privacy

A recent public opinion survey on AVs reported concerns over data privacy. Privacy concerns arise given that SAVs record all system operation and driving data, including related to crashes. As such, data-related privacy questions like the following arise: “Who should own and control the vehicle’s data? What types of data will be stored and how long? With whom will these data sets be shared? In what ways will such data be made available? And, for what ends will they be used?” [34]. These are all questions consumers may consider when purchasing an SAV.

### 5.2. Older Drivers Adapting to SAV

In order to have a safe deployment and transition toward increasingly capable SAVs, older adult drivers need to adopt, understand, trust, and properly use SAVs. Drivers’ knowledge and familiarity with SAV technology is poor [51]. Older adults are normally comfortable with new technology being integrated into their vehicles; however, there is some reluctance to relinquish control to an SAV believed to have inferior driving experience or that is not fully understood. In other words, older drivers are less confident, skeptical in these technologies, and thus least likely to trust and use them to improve their safety on the road [52,53]. Robertson et al. (2017) reported results of an online survey and focus groups, that drivers aged 70 years or older were less likely to feel safe using SAVs, related to feelings of poor knowledge about safety. Furthermore, a recent survey found that drivers are not adequately learning to use the systems, in part related to preferred method of learning [54]. As such, training for this type of technology will need to be adapted to the needs of older adult drivers so they can benefit from the range of capabilities SAVs have to offer and make them fit into their lifestyles and driving preferences. More research needs to be done about how this driving population acquire training and information related to new SAV technology [54]. Unfortunately, there may be a subset of older drivers where their cognitive decline actually precludes their ability to learn the new skills to adapt to SAVs.

In particular, the case of older adults with declining cognitive abilities will need to be considered if SAVs are to be considered as assistive devices for them. Even the adoption of partial autonomy will present challenges. Typically, older adults are slow to adopt new technology, including simple ones such as walkers and hearing aids, and these are further augmented in cases of cognitive decline. In line with our example above, autonomous vehicle makers will need to consider how an older adult with cognitive decline will react to a car not “allowing” them to change lanes because the signal light has not been activated. Driving automation will require additional training for older adults and more research will also need to be done on what older adults do in vehicles when automation is in control and how quickly they are able to re-take control [55,56]. Available learning strategies and improved training methods may allow drivers to modify their driving habits in response to traffic information and safety measures. For example, drivers may actually drive faster if they believe forward collision warning systems will improve their chances of stopping sooner. This behavioural adaptation could, in fact, negate the benefits of safety features in SAVs [51].

High levels of automation have been shown to result in driver complacency, caused by drivers becoming bored and fatigued, resulting in a decline in situational awareness; all of which can lead to mistakes and accidents [30]. Regression of driving ability is also a concern when drivers who have been “passengers” in a fully autonomous vehicle for years can no longer retake control if automation fails. Makers of autonomous vehicles will have to consider how these drivers will be able to take over the driving of the vehicle if its autonomy fails. Finally, there is the chance that drivers will take their SAV out of self-driving mode and assume control [57].

### 5.3. Adapting SAV to Older Drivers

Much work needs to be done for SAVs to become friendly for use by older adults. First of all, while young adults may like steering wheel vibration as a cue that they are crossing into another lane, that vibration may remind an older adult of a flat tire and cause them try to pull off the road. More research needs to be done as to how to format visual, auditory and/or sensory notifications for older adults. In addition, if future automobiles are to be truly older adult friendly, they will need to address those physical and cognitive deficits that affect older adults. The model presented in this paper could be used as a guide for SAV development, especially for the cognitive issues affecting driving associated with aging. Finally, designing a truly driver friendly SAV should include older drivers in a process of co-design.

## 6. Discussion

While SAVs are an exciting and rapidly evolving area of research that could potentially have positive impacts on traffic, safety, travel times, fuel consumption, and greenhouse gas emissions, there are many challenges to consider by automobile manufacturers to ensure SAV specifications meet the evolving cognitive functions of older adult drivers. Older drivers are the most likely parties to benefit from SAV technology given the changing cognitive capabilities related to visual scanning, attention, speed of processing, executive function, and memory. The possible negative impacts these cognitive changes have on actual driving whether it is lane drifting, suddenly stopping through moving traffic, or even getting lost, can be eradicated through the development of SAV features. Automation features to consider by automobile manufacturer included lane departure warnings, speedometer, blind spot detection, front/rear cross-check, navigation system, and intersection identification. In the future, these SAV features could be considered as assistive devices that some drivers require to delay driving retirement.

While this paper focused on cognitive changes, these automation features can also support older adults who experience reduced sensory abilities and physical function accompanied with aging. It is also important to consider that there can be neuropsychiatric issues associated with neurocognitive declines/syndromes, such as apathy, impulsivity, aggression, compromised decision-making, and lack of judgement/insight that can impact driving ability. Future automation may be developed to mitigate these as well.

The significant benefits SAVs bring forward for this growing population (e.g., independence, reduction in social isolation, and access to essential services) [43], will have to be considered when considering the challenges related to affordability, ethical liability, and privacy of such technology. There remain a number of issues that will need to be resolved, including the safety, design, and reliability of the technology. In addition, which notification mode older adults will respond to needs to be considered and how older adults will learn to adapt to the new technology will need to be researched. Indeed, to fully consider the impact of SAVs on older drivers experiencing cognitive changes, the latter could be directly involved in the design process.

Therefore, it is essential for automobile manufacturers to adopt SAV features to older adults’ cognitive needs. Understanding and learning opportunities targeting the proper usage of this technology can allow older drivers to overcome barriers faced with SAV technology. Consideration of our proposed model can support automation technology development to better meet the evolving cognitive and physical conditions of older drivers. 

Current vehicles are at autonomy levels 0 and 1. In addition to the technical work that needs to be done to get to higher levels, much more consideration needs to be given to the transition period leading to full automation. In particular, in higher levels of semi-automation, drivers are still required to be on stand-by, but their skills will diminish as the vehicles become “smarter.”

This paper develops a model that begins with cognitive elements associated with aging and their impact on driving ability and behaviour. It then looks for sensors that can assist with identifying these driving outcomes, and finally work can be done on developing vehicle responses. It is hoped that this model encourages collaborations between automobile manufacturers, academic computer engineers, and academic clinicians to help delay the driving retirement of older drivers.

## 7. Conclusions

Driving automation could become the assistive device of the future that delays driving retirement. Partial vehicle automation will continue to evolve over the next decades and needs to take the needs of older adults into consideration. Full automation for vehicles is still many years away and may change the requirements of driving ability altogether. Furthermore, disruptors like driverless vehicles could change vehicle ownership patterns, but will only happen with the advent of full automation. In addition to development of these technologies, efforts will need to be put into addressing the numerous ethical and legal challenges that will arise. There will need to be significant changes in public policy to regulate vehicle automation. The interdisciplinary collaboration of older drivers, clinicians, engineers, vehicle and automation manufacturers, and public policy makers will be required to advance the field of vehicle automation for it to become a safe assistive device in the future.

## Figures and Tables

**Figure 1 geriatrics-04-00063-f001:**
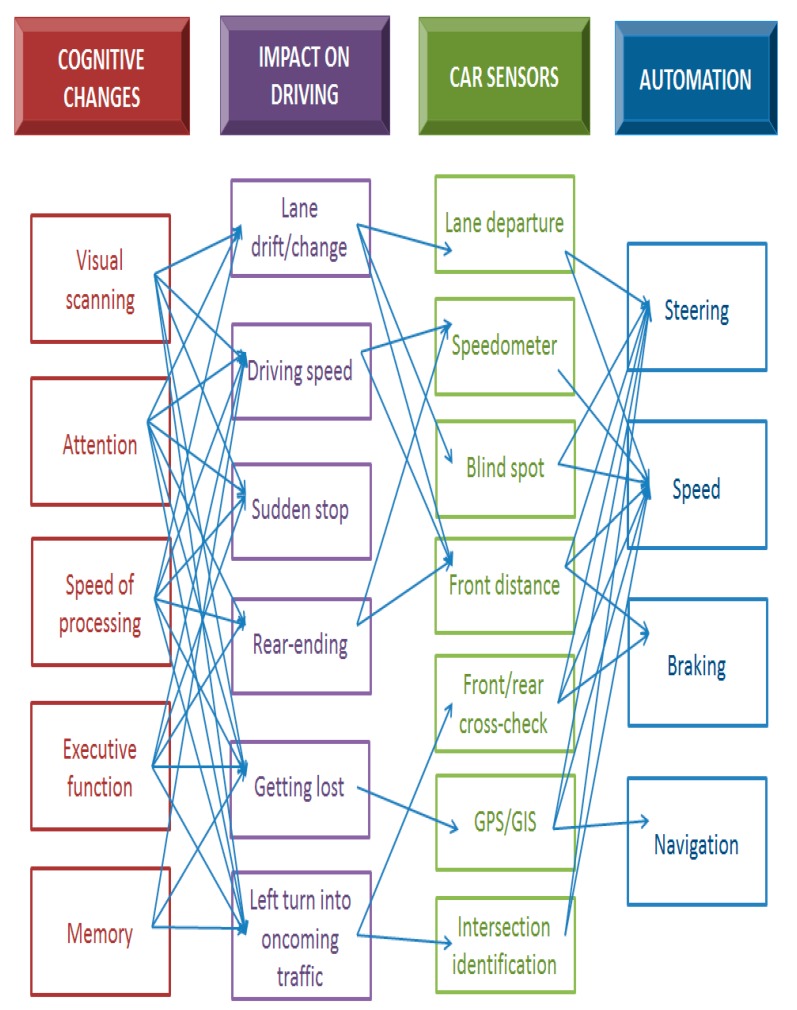
A complex model of connections between driving changes associated with cognitive decline and vehicle automation.

**Figure 2 geriatrics-04-00063-f002:**
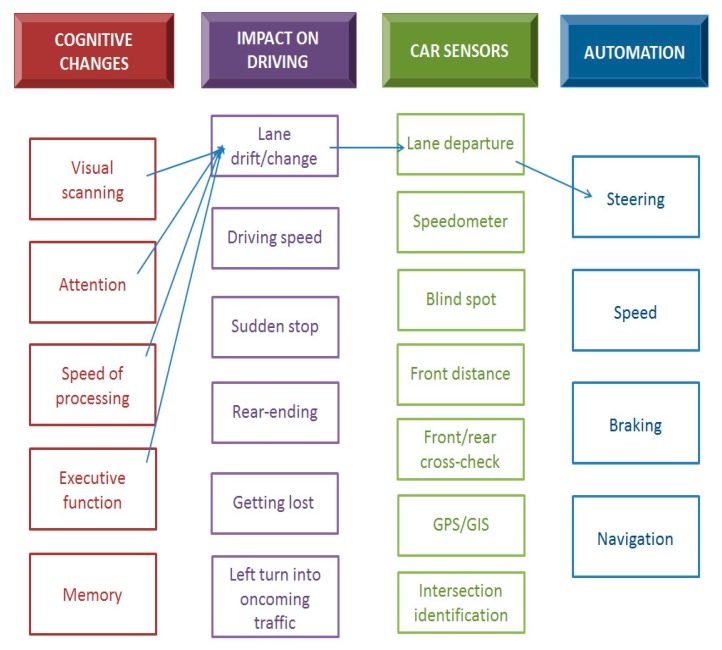
Example of cognitive changes impacting lane drift capacity.

**Table 1 geriatrics-04-00063-t001:** Vehicle levels of automation and features.

Level of Automation	Autonomous Features
Level 0: No automation	∙ Require constant human driver attention and control in every vehicle aspect (e.g., brakes, steering, and acceleration)
Level 1: Driver assistance	∙ Some assistance, but human driver attention and control required for fallback performance and environment monitoring. ∙ Vehicle responsible for some driving modes (e.g., emergency braking, blind spot detection, and/or lane keeping)
Level 2: Partial automation	∙ Require some human driver control and/or degree of situational awareness. ∙ Vehicle takes control over some aspects (e.g., steering and acceleration/deceleration.
Level 3: conditional automation	∙ Some driving modes are assumed by the vehicle and the vehicle is responsible for monitoring the environment. ∙ Human driver is required to be receptive to alerts, or other driving relevant system outputs, and respond if there is a request to intervene.
Level 4: High automation	∙ Vehicle in control of specific modes of autonomous operations and/or within a specified area (Operational Design Domain—ODD) even if the human driver does not respond to a request to intervene. ∙ Human driver is not expected to intervene. Requests to intervene are still possible, but fallback performance now lies with the driver, which means that in case of an emergency, or if the request to intervene is not responded to, the vehicle automatically assumes a minimal risk condition.
Level 5: Full automation	∙ Human driver has no responsibility for monitoring the environment. They can request the vehicle to attain a minimal risk condition. ∙ All driving aspects and fallback performance are assumed by the vehicle. Requests to intervene are still possible.

**Table 2 geriatrics-04-00063-t002:** Automotive sensing categories.

Sensing Category	Description
Self-sensing	Vehicle uses proprioceptive sensors such as pre-installed measurement units (e.g., odometers, inertial measurement units (IMUs), gyroscopes, and controller area network (CAN) bus) to measure the current state of the ego-vehicle, including the vehicle’s wheel velocity, acceleration, rotational velocity, yaw, and steering angle.
Localization	Vehicle uses external sensors such as GPS or dead reckoning by IMU readings to determine the vehicle’s global and local position.
Surroundingsensing	Vehicle uses exteroceptive sensors to detect road markings, road slope, traffic signs, weather conditions, the state (position, velocity, acceleration, etc.) of obstacles including other vehicles, and even the state of the driver (vigilance, drowsiness, fatigue, boredom due to monotony, etc.).

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
