# Peer review of "Semi-Autonomous Vehicles as a Cognitive Assistive Device for Older Adults"

_geriatrics, 2019, doi:10.3390/geriatrics4040063_

Round 1

Reviewer 1 Report

Thank you for the opportunity to review this paper.  

It was nicely written and easy to read.  However, I am not sure I actually learned anything new.  For those in this research area, there is little of novel information.  The proposed model is interesting, but once you go past some simple ideas, it becomes messy.  However, that is the point - there is not an easy solution, or even a solution at this point.

However, while what I was reading was things I already know, the paper did a nice job of summarizing the current state of affairs.  SAVs are not going to "save the day," as there as too many issues.  Again, nothing particularly new, but nicely summarized.

I do think the discussion section - lines 364-369 are redundant with the introduction of the paper (impact of aging on driving)  Do you really need a discussion? It is not really a research paper and each section essentially is a discussion about the issues.  I was loosing interest fast in the discussion section.  Do the summary and be done.

Get rid of "cool".

Author Response

REVIEWER 1:

Thank you for the opportunity to review this paper. It was nicely written and easy to read. However, I am not sure I actually learned anything new. For those in this research area, there is little of novel information. The proposed model is interesting, but once you go past some simple ideas, it becomes messy. However, that is the point - there is not an easy solution or even a solution at this point. However, while what I was reading was things I already know, the paper did a nice job of summarizing the current state of affairs. SAVs are not going to "save the day," as there as too many issues. Again, nothing particularly new, but nicely summarized.

We thank reviewer 1 for their review and for their thoughtful comments.

I do think the discussion section - lines 364-369 are redundant with the introduction of the paper (impact of aging on driving) Do you really need a discussion? It is not really a research paper and each section essentially is a discussion about the issues. I was losing interest fast in the discussion section. Do the summary and be done.

We removed a significant portion of the discussion as recommended. We left parts that suggest that the paper is not covering all cognitive, psychological and physical aspects of aging – for clarity.

Get rid of "cool".

We replaced the sentence with “cool” to:  “While SAVs are an exciting, and rapidly evolving area of research…”

Reviewer 2 Report

This is an interesting and timely review of aging and semi-autonomous vehicles (SAV). The authors have done a good job of bringing together the relevant literature and to discuss the important issues related to SAV.

In the introduction, the authors are a bit too negative and pessimistic about the cognitive and driving performance of older subjects. In addition, they use the term "executive function" a bit too sloppy.

I would also refer a bit to the beneficial effects of training and here on-road and driving simulator training on driving performance (e.g., Casutt et al., 2014a).

Page 1, Lines 40 following: "There is general agreement that with aging there is a complex interaction of genetics, life
41 experience, and lives ...". This statement needs several references. In addition, it is not clear whether the mentioned physiological, anatomical and psychological aspects are really correlated with each other (e.g., Oschwald et al., 2019).

The authors mention that "Executive functioning is the brain’s most complex function, and includes elements such as dividing attention, cognitive flexibility, inhibition, planning, reasoning, and self-regulation." Is this common sense or is this just a narrative statement. Executive functions are a plethora of psychological functions, thus it is not a unified psychological function. And why are EF the most complicated?

On page 2, they also mention "Driving research has shown that decline in executive functioning reduces the speed at which visual information and attention is processed when reacting between two aspects of driving in potentially difficult or dangerous traffic situations." See above for the theoretical problem.

Although several studies report correlations between psychological test performances and driving performances, these correlations are often very small or sometimes even absent (e.g., Casutt et al., 2014b). Maybe the authors to discuss that a bit.

In general a fine and important paper!

References

Casutt, G., Theill, N., Martin, M., Keller, M., & Jäncke, L. (2014a) The drive-wise project: driving simulator training increases real driving performance in healthy older drivers. Front. Aging Neurosci., 6, 85.

Casutt, G., Martin, M., Keller, M., & Jäncke, L. (2014b) The relation between performance in on-road driving, cognitive screening and driving simulator in older healthy drivers. Transp. Res. Part F Traffic Psychol. Behav., 22, 232–244.

Oschwald, J., Guye, S., Liem, F., Rast, P., Willis, S., Röcke, C., Jäncke, L., Martin, M., & Mérillat, S. (2019) Brain structure and cognitive ability in healthy aging: a review on longitudinal correlated change. Rev. Neurosci.,.

Author Response

 REVIEWER 2: 

This is an interesting and timely review of aging and semiautonomous vehicles (SAV). The authors have done a good job of bringing together the relevant literature and to discuss the important issues related to SAV.

We thank reviewer 2 for their review and thoughtful comments. 

In the introduction, the authors are a bit too negative and pessimistic about the cognitive and driving performance of older subjects. 

We tried to decrease the negative tone of the introduction by softening the sentence “age-related declines… can potentially affect older adults on the road.” We also added “Given all of these cognitive changes… potentially every driver would develop changes sufficient to increase crash rate.”

In addition, they use the term "executive function" a bit too sloppy.

I would also refer a bit to the beneficial effects of training and here on-road and driving simulator training on driving performance (e.g., Casutt et al., 2014a).

We also included the benefits of different training methods and cited the Casutt et al., 2014a study in the first section.

Page 1, Lines 40 following: "There is general agreement that with aging there is a complex interaction of genetics, life 41 experience, and lives ...". This statement needs several references.

In addition, it is not clear whether the mentioned physiological, anatomical and psychological aspects are really correlated with each other (e.g., Oschwald et al., 2019).

Regarding the section on aging we included additional references as requested: Passarino et al (2016), Roelfsema et al (2018) and included the Oschwald et al (2019) reference. 

The authors mention that "Executive functioning is the brain’s most complex function, and includes elements such as dividing attention, cognitive flexibility, inhibition, planning, reasoning, and self-regulation." Is this common sense or is this just a narrative statement. Executive functions are a plethora of psychological functions, thus it is not a unified psychological function. And why are EF the most complicated?

On page 2, they also mention "Driving research has shown that decline in executive functioning reduces the speed at which visual information and attention is processed when reacting between two aspects of driving in potentially difficult or dangerous traffic situations." See above for the theoretical problem. 

We used the reviewer’s definition of “executive functioning,” and fixed the sentence that mixed executive functioning and visual information: 

“Executive functions are a plethora of psychological functions that include elements such as dividing attention, cognitive flexibility, inhibition, planning, reasoning, and self-regulation….”

“Driving research has shown that a decline in executive functioning can include a reduction in the speed at which visual information is processed when switching between two aspects of driving, and that this can lead to potentially dangerous traffic situations.”  

Although several studies report correlations between psychological test performances and driving performances, these correlations are often very small or sometimes even absent (e.g., Casutt et al., 2014b). Maybe the authors to discuss that a bit.

We included a statement indicating the weak relationship between cognitive tests and actual on road performance and included the recommended reference as well as another more recent review. 

In general a fine and important paper!

Thank you!

Reviewer 3 Report

An important area of research for future practice. Well done.

This is an important and useful paper that contributes to knowledge and future applications to explore what technological developments will assist older drivers abilities to safely drive.
I recommend a rework to aid the clarity of the message and increase the rigour of being embedded in literature.
Introduction page 2
In terms of cognitive domains justify why these in particular have been selected – just needs a statement about research identifying they relate to driving performance in older people and reference
line 48- visual field ‘try to’ compare what is being seen to the []“ information being looked at” Reword for academic writing and increase clarity.
Line 61- coming to the brain- paraphrase perhaps carried into the brain via ….
Line 73 add for example a child on a bicycle
Page 3- line 105- grammar difficulty in navigating- do not need to add get lost sufficient reaction time to avoid collisions
Line 111- would- consider potentially as not all older drivers have this functional decline that influence driving
Line 118- driver at [an} increased
Line 120 rather than around- others including family
Line 125- 137- rather than list include in text synthesising this information in a paragraph
Page 4- the spacing in the Table is not correct as levels are not consistently matched to autonomous vehicles as with Table 2
Page 6 the model- this is interesting but messy. Consider just focussing on Impact on driving – but change wording to functional impact on driving or refer to paper by reference below related to ICF and driving related to driving which may put in more universal language for different professional groups to understand.
George, S., May, E., & Crotty, M. (2009). Exploration of the links between concepts of theoretical driving models and the international classification of functioning, disability, and health. Journal of allied health, 38(2), 113-120.
In terms of cognitive domains justify why these in particular have been selected – just needs a statement about research identifying they relate to driving performance in older people and reference

The arrows are messy so consider a table you list functional change and next column the cognitive changes and only do arrows to car sensors and automation which are much clearer as less numerous
Figure 2 is clearer
Page 10- line 349 “works to the right” reword to increase clarity ie an understanding of …. Specific Older adults cognitive changes …,
You mention physical sometimes so consider incorporating in intro and adding throughout.as only mentioned from page 10 on.
Line 382 rather than seriously- just considered
The other factor that need to include is line 386 re adapting to new technology also is how people will learn to use when have cognitive changes- needs to be inherent in design – consider concepts of co-design with end users

Author Response

REVIEWER 3: 

We thank reviewer 3 for their review and thoughtful comments.

This is an important and useful paper that contributes to knowledge and future applications to explore what technological developments will assist older drivers’ abilities to safely drive.  I recommend a rework to aid the clarity of the message and increase the rigor of being embedded in literature.  Introduction page 2 In terms of cognitive domains justify why these in particular have been selected – just needs a statement about research identifying they relate to driving performance in older people and reference 

We added a statement to clarify why these specific domains were selected and included two references:

“Many cognitive domains can be impacted by aging, but for the purpose of creating a model, this paper will focus on visual scanning, attention, speed of processing, some components of executive functioning and memory, as these domains have repeatedly been shown to be related specifically to driving performance in older adults (5,6).”

line 48- visual field ‘try to’ compare what is being seen to the []“information being looked at” Reword for academic writing and increase clarity. 

We removed this sentence.

Line 61- coming to the brain- paraphrase perhaps carried into the brain via ….

We changed the sentence as suggested.

Line 73 add for example a child on a bicycle

We added “for example a child on a bicycle.”

Page 3- line 105- grammar difficulty in navigating- do not need to add get lost sufficient reaction time to avoid collisions

We removed these sections as recommended.

Line 111- would- consider potentially as not all older drivers have this functional decline that influence driving

We added “potentially” as suggested.

Line 118- driver at [an} increased

We added “an” as suggested.

Line 120 rather than around- others including family

We made the changes as suggested.

Line 125- 137- rather than list include in text synthesizing this information in a paragraph

We made this change as suggested.

Page 4- the spacing in the Table is not correct as levels are not consistently matched to autonomous vehicles as with Table 2

We adjusted the alignment of the text in Tables 1 and 2.

Page 6 the model- this is interesting but messy. Consider just focusing on Impact on driving – but change wording to functional impact on driving or refer to paper by reference below related to ICF and driving related to driving which may put in more universal language for different professional groups to understand.

Thank you for this suggestion, but part of the purpose of this paper is to tie together these issues into one “messy” picture. 

George, S., May, E., & Crotty, M. (2009). Exploration of the links between concepts of theoretical driving models and the international classification of functioning, disability, and health. Journal of allied health, 38(2), 113-120.

Thank you for this reference, but we could not find a good place to refer to it in this paper.

In terms of cognitive domains justify why these in particular have been selected – just needs a statement about research identifying they relate to driving performance in older people and reference

We included a reference as above.

The arrows are messy so consider a table you list functional change and next column the cognitive changes and only do arrows to car sensors and automation which are much clearer as less numerous Figure 2 is clearer

Thank you for this comment. Indeed, part of the purpose of having two tables was to show the complexity of the interactions in the first table and then to dissect out one simple, clear example in the second. We will ensure that a high-quality image is used in the publication for visual clarity. 

Page 10- line 349 “works to the right” reword to increase clarity ie an understanding of …. Specific Older adults cognitive changes …,

We rewrote this sentence as follows: “The model presented in this paper could be used as a guide for SAV development, especially for the cognitive issues affecting driving associated with aging.”

You mention physical sometimes so consider incorporating in intro and adding throughout.as only mentioned from page 10 on.

We did not want to complicate the paper further, so focused this paper on cognition. Addressing how physical challenges could be impacted by vehicle autonomy could be a new paper?

Line 382 rather than seriously- just considered

We removed “seriously.” As recommended.

The other factor that need to include is line 386 re adapting to new technology also is how people will learn to use when have cognitive changes- needs to be inherent in design – consider concepts of co-design with end users

We added a statement to this effect as recommended.

Reviewer 4 Report

A very thoughtful piece. My comments below:

Should be flagged earlier in introduction that ethical and legal issues will be discussed later. Neuropsychiatric issues associated with neurocognitive declines/syndromes, such as apathy, impulsivity, aggression, compromised decision-making, and lack of judgement/insight, are absent, and should be addressed fully, in addition to the cognitive changes discussed, otherwise the complete picture of brain function in driving is incomplete. These factors should be addressed within the article as well as in the discussion. Section 5.1.3 should be expanded with respect to ethical issues e.g. primary care physician vs family vs patient disagreeing on ability of patient to drive? Who can be sued? What about the emotional costs of an older driver in a SAV who kills their grandchildren? These potential negative outcomes of continuing to drive with a SAV should be weighed against potential negative effects of driving cessation (I refer specifically to Lines 128-137 in the manuscript). Ethical/legal concerns should also be mentioned in the final conclusion section. In section 5.2 driver training is addressed; what if cognitive issues preclude a level of adequate training to use the SAV? Also, older drivers may be uncomfortable with the technology because they have not, as a user group, been meaningfully involved in the development of this technology (see lines 389-390 in Discussion - here the co-design of SAV systems between designers, researchers and older drivers could be mentioned as potentially leading to both design insights as well as improved acceptance/adaptation to SAVs).

MINOR ISSUES:

3. Line 292: Sentence beginning "For example..." is a fragment.

Author Response

REVIEWER 4:

We thank reviewer 4 for their review and thoughtful comments.

A very thoughtful piece. My comments below:  Should be flagged earlier in introduction that ethical and legal issues will be discussed later. Neuropsychiatric issues associated with neurocognitive declines/syndromes, such as apathy, impulsivity, aggression, compromised decision-making, and lack of judgement/insight, are absent, and should be addressed fully, in addition to the cognitive changes discussed, otherwise the complete picture of brain function in driving is incomplete. These factors should be addressed within the article as well as in the discussion. 

We added the following sentence regarding ethical and legal issues to the introduction: “The legal and ethical issues around driving retirement or its delay are significant.

We agree with the reviewer that neuro-psychiatric issues associated with neurocognitive decline are important and that they can have an important impact on driving ability. However, we are not aware of any current driving automations that can mitigate them. We added a statement to the discussion to this effect:

It is also important to consider that there can be neuropsychiatric issues associated with neurocognitive declines/syndromes, such as apathy, impulsivity, aggression, compromised decision-making, and lack of judgement/insight, that can impact driving ability. Future automation may be developed to mitigate these as well. 

Section 5.1.3 should be expanded with respect to ethical issues e.g. primary care physician vs family vs patient disagreeing on ability of patient to drive? Who can be sued? What about the emotional costs of an older driver in a SAV who kills their grandchildren? These potential negative outcomes of continuing to drive with a SAV should be weighed against potential negative effects of driving cessation (I refer specifically to Lines 128-137 in the manuscript). 

We added the following to section 5.1.3:

Additional legal issues can arise if there are disagreements between physicians, families and older drivers regarding the need of driving automation – if there is an accident that could have been avoided using automation, who is responsible? Similarly, if there is an accident despite automation, who is legally responsible: the physician that “prescribed” it, the driver, or the manufacturer? All of these issues will need to be taken into consideration when discussing with patients and families the risks and benefits when choosing to continue to drive with some driving automation vs. driving retirement.“

Ethical/legal concerns should also be mentioned in the final conclusion section. 

We added this to the conclusion:

In addition to development of these technologies, effort will need to be put into addressing the numerous ethical and legal challenges that will arise.

In section 5.2 driver training is addressed; what if cognitive issues preclude a level of adequate training to use the SAV? 

We added the following sentence to 5.2:

Unfortunately, there may be a subset of older drivers where their cognitive decline actually precludes their ability to learn the new skills to adapt to SAVs.”

Also, older drivers may be uncomfortable with the technology because they have not, as a user group, been meaningfully involved in the development of this technology (see lines 389-390 in Discussion - here the co-design of SAV systems between designers, researchers and older drivers could be mentioned as potentially leading to both design insights as well as improved acceptance/adaptation to SAVs). 

We added the following to section 5.3:

Finally, to design a truly older driver friendly SAV should include older drivers in a process of co-design.”

And changed the discussion to included: “Indeed, to fully consider the impact of SAVs on older drivers experiencing cognitive changes, the latter could be directly involved in the design process.” 

MINOR ISSUES: Line 292: Sentence beginning "For example..." is a fragment.

We removed “for example” and replaced it with “An example of such a situation could be…”

Round 2

Reviewer 4 Report

Thank you for the attention to corrections.